# A Novel, Circulating Tumor Cell Enrichment Method Reduces ARv7 False Positivity in Patients with Castration-Resistant Prostate Cancer

**DOI:** 10.3390/diagnostics10030151

**Published:** 2020-03-11

**Authors:** Takehiko Nakasato, Chiho Kusaka, Mika Ota, Yuki Hasebe, Kumiko Ueda, Tsutomu Unoki, Kazuhiko Oshinomi, Jun Morita, Yoshiko Maeda, Takeshi Shichijo, Michio Naoe, Yoshio Ogawa

**Affiliations:** Department of Urology, School of Medicine, Showa University, 1-5-8, Hatanodai, Shinagawa-ku, Tokyo 142-8666, Japan; bun_093@yahoo.co.jp (C.K.); mika.ohta18@gmail.com (M.O.); y.hasebe@med.showa-u.ac.jp (Y.H.); kueda@med.showa-u.ac.jp (K.U.); t-unoki@med.showa-u.ac.jp (T.U.); oshikazu@med.showa-u.ac.jp (K.O.); moritajun@med.showa-u.ac.jp (J.M.); ymaeda@med.showa-u.ac.jp (Y.M.); shichijo@med.showa-u.ac.jp (T.S.); naoemichio@med.showa-u.ac.jp (M.N.); ogawayos@med.showa-u.ac.jp (Y.O.)

**Keywords:** AR-V7, androgen receptors, docetaxel, cabazitaxel, abiraterone, enzalutamide

## Abstract

Background: The AR-V7 splice variant is a cause of castration-resistant prostate cancer (CRPC). However, testing for the presence of AR-V7 by real-time polymerase chain reaction (RT-PCR) shows AR-V7 positivity in healthy individuals. We hypothesized that the positivity reflects contamination by hematopoietic cells. We tried a novel circulating tumor cell (CTC) enrichment instrument, using Celsee, to clear hematopoietic cells. Methods: We tested whole blood or Celsee-enriched samples for AR-V7 by RT-PCR, and included samples from 41 CRPC patients undergoing sequential therapy. We evaluated the associations between AR-V7 status and clinical factors. We evaluated factors affecting AR-V7 positivity. Results: AR-V7 positivity was lower in Celsee-enriched than in whole blood specimens. AR-V7 and clinical factors did not predict the therapy effectiveness. We found no significant differences in the effectiveness of enzalutamide/abiraterone (Enz/Abi) upon AR-V7 evaluation. All AR-V7 positive patients had resistance to Enz/Abi. Docetaxel (DTX), cabazitaxel (CBZ), and Radium223 treatment showed no significant difference in the treatment effectiveness, regardless of AR-V7 presence. AR-V7 was more frequently positive than Extent of disease (EOD) 2 in cases with bone metastases. Conclusion: Celsee CTC enrichment suppresses AR-V7 false positivity. All AR-V7 positive patients presented resistance to Enz/Abi. DTX, CBZ, and Radium223 were effective and remain treatment options. AR-V7 positivity should progressively appear in patients with advanced bone metastases.

## 1. Introduction

Androgen receptors (ARs) are encoded by eight exons (exons 2–3 are DNA binding domains and exons 4–8 are ligand binding domains (LBDs)). The AR splice variant 7 (AR-V7) has exon 3 followed by a cryptic exon 3 and omits exons 4 to 8. It lacks the LBD but retains functional, transcriptive element binding domains that mediate intracellular AR signaling in a ligand-independent manner [1,2].

AR-V7 has attracted attention because of its association with castration-resistant prostate cancer (CRPC). Enzalutamide (Enz), an inhibitor of AR signaling that binds to the LBD of the AR [3,4], and abiraterone (Abi), an inhibitor of cytochrome P450 17A1 that impairs AR signaling [5,6], are the main androgen axis drugs against CRPC. After analyzing AR-V7 in circulating tumor cells (CTCs), Antonarakis reported that patients with AR-V7-positive CTCs presented high resistance to Enz and Abi [7]. Alternatively, docetaxel (DTX) and cabazitaxel (CBZ), which are taxane preparations, showed some promise for patients with AR-V7-positive tumor cells [8,9]. Thus, AR-V7 has been considered a biomarker to guide treatment choices. Bernemann found that about 20% of patients expressing AR-V7 in CTCs had a prostate-specific antigen (PSA) reduction of more than 50% after treatment with Enz and Abi [10], and the PROPHECY study found that the effectiveness of Enz and Abi reached 6–11% of AR-V7 positive patients when the Johns Hopkins Hospital (JHU) method by Antonarakis was used to assess the presence of the biomarker. These results point to a diagnostic inaccuracy of the biomarker. Alternatively, the effectiveness of Enz and Abi was 0% in AR-V7 positive patients when using the Epic Sciences platform to assess the presence of the marker [11]. The positivity of JHU AR-V7 and Epic AR-V7 may vary even in results from a single patient, which indicates that CTC analysis differences may influence the accuracy of AR-V7 as a biomarker.

CTC analysis is usually performed in two steps (CTC enrichment and detection), but no standardized method exists. Problems during CTC enrichment include the possibility of contamination by hematopoietic cells and missing CTCs. The PCR detection is directly affected by the accuracy of CTC enrichment, and its results may reflect contaminations by cells other than CTCs [12]. For example, whole blood can be stabilized and stored with minimal gene degradation using PAXgene (Preanalytix, Hombrechtikon, Switzerland), but when using samples without CTC enrichment, 75.7% of individuals without prostate cancer test positive for AR-V7, reflecting a low specificity for the test system [13]. Hematopoietic cells expressed prototype AR mRNAs [14], and they may also express AR-V7 (for example, in leukocytes) [15]. In a most recent report, there have been reports of AR-V7 expression in blood cells. Basal levels of AR-V7 in the non-cancer population collected peripheral blood mononuclear cells (PBMC) from 24 non-cancer individuals. AR-V7 was detected in 18 of them (75%) [16].

Given the differing sensitivity and specificity of AR-V7 assessment tests, depending on the test system, we considered the following two points to be important when assessing AR-V7 positivity: one was to exclude hematopoietic cells as much as possible during CTC enrichment, the second one was to target AR-V7 mRNA from CTCs during detection.

Celsee (Celsee Diagnostics, Plymouth, MI, USA) is a microfluidic device that can be used for CTC enrichment. Compared with CellSearch, which is the only system approved by the U.S. Food and Drug Administration (FDA), the capture rate of Celsee for CTCs is high [17]. The basic principle is based on the larger size and non-deformability of CTCs compared with those of hematopoietic cells; thus, the system’s chamber ensures that small hematopoietic cells escape, whereas larger CTCs get trapped and can be isolated in the chamber. The Celsee microfluidic device allows for physical CTC enrichment; its CTC capturing efficiency is greater than 80% and the background of hematopoietic cell contamination in the captured cell population is minimal [18]. For CTC identification, we used real-time polymerase chain reaction (RT-PCR) to detect AR-V7 messenger ribonucleic acid (mRNA).

We hypothesized that Celsee CTC enrichment would suppress AR-V7 false positives. The first aim of this study was to confirm the difference in the detection rate of AR-V7 between the PAXgene systems without CTC enrichment and those with Celsee CTC enrichment. We used samples from patients with CRPC treated with sequential therapy.

We assumed that analysis of AR-V7 using Celsee would predict the outcome of CRPC treatment, and our second aim was to evaluate the associations between the current treatment efficacy and variables, such as the presence of AR-V7, and prognostic or clinical factors (PSA levels, alkaline phosphatase (ALP), lactate dehydrogenase (LDH), albumin (Alb), hemoglobin (Hb), metastatic sites on viscera, bone, or lymph nodes, Gleason score (GS), time-to-CRPC (period from the start of hormone therapy to CRPC), time-from-CRPC (period since becoming CRPC), and the history of therapeutic drug use).

AR-V7 is thought to be positive during CRPC treatment. Our third aim was to identify factors affecting AR-V7 positivity by examining the association between prognostic and clinical factors.

## 2. Materials and Methods

### 2.1. Patients

From January 2018 to September 2019, the study included a total of 41 patients with CRPC, including 38 with metastatic CRPCs (mCRPC) and three with non-metastatic CRPC (nmCRPC). All patients were treated with surgical castration, degarelix acetate, leuprorelin acetate, or goserelin acetate as androgen deprivation therapy in the Showa University hospital and received CRPC treatments with Enz, Abi, DTX, CBZ, Radium223, and traditional hormones (bicalutamide, flutamide, estramustine, phosphate sodium hydrate, and ethinyl estradiol). We also included samples from eight healthy controls recruited from volunteers without prostate cancer in the Showa University Urological Department. The Showa university clinical research ethics boards approved the study (approval number 2325: 15 January 2018). All patients provided written informed consents.

### 2.2. Study Design

We measured PSA levels and obtained other laboratory tests for monitoring monthly. Imaging such as computed tomography (CT), magnetic resonance imaging (MRI), and bone scintigraphy were performed at least every 3–6 months.

We defined stable disease as that in individuals with a <50% PSA decrease to <25% PSA increase from a PSA nadir, and disease response as that in individuals with a PSA decrease >50% in the absence of radiographic progression. We defined disease progression as radiographic progression or PSA increases ≥25% and ≥2 ng/mL above the nadir or baseline. We used Prostate Cancer Clinical Trials Working Group (PCWG) 3 soft tissue and bone scan criteria to assess radiographic progression [19]. Time points for taking peripheral blood samples for analysis of AR-V7 were performed on patients already undergoing treatment for CRPC. Treatment selection was at the discretion of the treating physician, without AR-V7 status consideration. Laboratory investigators were blinded to the clinical information and patient outcomes.

### 2.3. Blood Collection and RNA Extraction

#### 2.3.1. Using Celsee

We placed 5 mL of blood in anticoagulant, ethylene–diamine–tetra–acetic acid (EDTA) blood tubes. We treated 4 mL blood samples with red blood cell (RBC) lysis buffer (10×) (BioLegend, San Diego, CA, USA) and let them stand for 20 min on ice. Following centrifugation at 326× *g* for 5 min at room temperature, we discarded the supernatants and resuspended the isolated cells in 8 mL of 1% Bovine Serum Albumin (BSA)/ Phosphate-Buffered Saline (PBS). We ran the samples through Celsee and collected the cells on the chip in 1 mL × 8 back flushes using 1% BSA/PBS, and centrifuged them at 326× *g* for 5 min. We used an RNA isolation NucleoSpin kit (MACHEREY-NAGEL, Düren, Germany), according to the manufacturer’s protocol.

#### 2.3.2. Using Paxgene

We collected 2.5 mL blood samples into Paxgene blood tubes and stored them at −80 °C. We thawed the blood samples prior to extracting RNA from them using the Paxgene blood RNA kit (Preanalytix, Hombrechtikon, Switzerland), according to the manufacturer’s protocol.

### 2.4. RNA Quality Control and Reverse Transcription

We processed all samples after RNA extraction with a DNase kit (Turbo-DNA-free AM1907; Invitrogen, CA, USA) and then ethanol-precipitated the DNA-free RNA samples with Ethachinmate (NIPPON GENE, Japan).

We used the NanoDrop One^C^ system to determine RNA concentrations and absorption at 260 and 280 nm. We only used samples with a 260/280 ratios ≥2.0. We performed reverse transcription (RT) with 1 µg of RNA using a high-capacity complementary DNA (cDNA) Reverse Transcription Kit (Applied Biosystems, Waltham, MA, USA), according to the manufacturer’s protocol.

### 2.5. Polymerase Chain Reaction

We used the SYBR Green method for our RT-PCR assays. We made cDNA using the KAPA SYBR FAST qPCR Master Mix (KAPA Biosystems, Headquarters, MA, USA) to detect prostate cancer-associated RNA transcripts.

We adapted the tests for detection of AR-V7 by RT-PCR, using custom primers specific for AR-V7 (forward: 5′-CCATCTTGTCGTCTTCGGAAATGTTA-3′, reverse: 5′-TTTGAATGAGGCAAGTCA GCCTTTCT-3′). We used primers for glyceraldehyde-3-phosphate dehydrogenase (GAPDH) (forward: 5’-GCACCGTCAAGGCTGAGAAC-3’, reverse: 5′-TGGTGAAGACGCCAGTGGA-3′) as a housekeeping gene.

We performed RT-PCR under optimized conditions at 95 °C × 30 s, 95 °C × 5 s, and 60 °C × 30 s for 40 cycles, as well as 95 °C × 10 s and 65 °C × 5 s followed by melting curve analysis.

We measured product concentrations using the CFX96 (Bio-Rad, Hercules, CA, USA) instrument and analyzed results with the CFX Maestro software (Bio-Rad). We confirmed that all AR-V7 positive specimens were GAPDH-positive.

### 2.6. Basic Experiments on Celsee Before Clinical Trials

We collected 4 mL of whole blood samples from eight healthy volunteers into EDTA blood tubes. Subsequently, we performed RNA extraction, RT, and PCR assays on these specimens, and confirmed AR-V7 positivity in six cases (Figure 1).

The figure shows the results of PCR tests in AR-V7-positive reactions of five healthy volunteers and Milli-Q water as a negative control (green line). The melting peak of AR-V7 lies around 0.5 degrees of 81.5 degrees.

We observed an increase in Threshold Cycle (Ct) value with Milli-Q water, which was used as a negative control, but this probably reflects amplification of primer dimers. We considered AR-V7 positivity only when these conditions applied simultaneously: the melting peak was the same as that for AR-V7, and the Ct value increased.

We found a Ct value elevation in the negative control (green line), but the melting peak clearly differed from it. We confirmed AR-V7 positivity in samples from six of eight individuals.

We passed 4 mL blood samples from the six healthy volunteers positive for AR-V7 into the whole blood analysis through a Celsee instrument. The RT-PCR results confirmed these samples were negative for AR-V7 (Figure 2).

We used Vcap as a positive control (red line) and Milli-Q water as a negative control (green line). The other two lines (blue and brown) are those for the healthy volunteers’ samples.

The figure shows that the whole blood of the two healthy volunteers became negative after Celsee enrichment, and the melting peak was detected on the primer dimer side, but not on the AR-V7 graph line. We were able to confirm similar results for all six patients who displayed AR-V7 positivity in their whole blood samples.

We spiked 10 and 100 Vcap (ATCC CRL-2876) into 4 mL healthy volunteer blood samples and added Celsee enrichment in the same manner to detect AR-V7 in RT-PCR. We confirmed AR-V7 detection (Figure 3).

The red line in Figure 3 indicates the Vcap positive control, the green line indicates the Vcap 0 spike sample, the blue line indicates the Vcap 10 spike sample, and the pink line indicates the Vcap 100 spike sample.

We found a melting peak similar to that of AR-V7 in the Vcap (10 and 100) spike group, and confirmed the melting peak only in the primer dimer graph portion in the Vcap 0 spike.

By these procedures, we showed that the AR-V7 positivity in the whole blood of healthy volunteers became negative after Celsee enrichment; thereby, we confirmed that tumor-derived AR-V7 can be detected.

By these procedures, the AR-V7 positivity in the whole blood samples of healthy volunteers became negative, and we confirmed that tumor-derived AR-V7 cells can be detected.

### 2.7. Data Analysis

We did not detect AR-V7 after Celsee enrichment in healthy individuals, and thus, we did not set a Ct value threshold for it in this study. We defined AR-V7 positivity as when GAPDH was positive by PCR and the melt peak of AR-V7 was confirmed within 0.5 degrees around 81.5.

We performed all statistical analyses using the SPSS Software v.21.0 (SPSS, Chicago, IL, USA). We applied the Mann–Whitney U test to compare continuous variables (e.g., age and PSA), and Fisher’s exact test for categorical variables (e.g., AR-V7 positivity and metastasis). We further analyzed factors with a significant difference using a multivariate logistic regression analysis. We considered *p* < 0.05 as statistically significant.

## 3. Results

Table 1 lists the baseline characteristics of 41 patients with CRPC. We found AR-V7 positivity in 22 (53.7%) PAXgene-processed samples and in nine (22.0%) Celsee-processed ones. We confirmed AR-V7 positivity differences between PAXgene and Celsee-processed samples from single patients.

Table 2 shows the factors affecting the effectiveness of the treatments on univariate analysis. PSA (SD) was 45.2 ng/mL (108.9) for stable disease or response, and 370.0 ng/mL (508.2) for disease progression (*p* = 0.016). **□** Hemoglobin (Hb) (SD) was 12.1 g/dl (1.4) for stable disease or response and 10.6 g/dl (1.5) for disease progression (*p* = 0.005). The time-to-CRPC (SD) was 36.9 months (35.6) for stable disease or response and 16.7 months (13.6) for disease progression (*p* = 0.005). 

Table 3 shows the results of our multivariate analysis. We found no significant differences in terms of PSA (Hezard Ratio (HR) = 1.004; 95% confidence interval (CI) = 0.999–1.009; *p* = 0.143), Hb (HR = 0.651; 95% CI = 0.374–1.135; *p* = 0.13), or time-to-CRPC (HR = 0.958; 95% CI = 0.907–1.012; *p* = 0.125).

We confirmed the effectiveness of Enz and Abi in past and current treatments of patients with or without AR-V7 positivity. We found no statistically significant differences, but none of the AR-V7-positive patients displayed Enz and Abi treatment effectiveness (*p* = 0.066; Table 4).

We examined the effectiveness of DTX, CBZ, and Radium223 with and without AR-V7 positivity, and found no statistically significant differences between treatments (*p* = 0.217; Table 5).

Table 6 shows factors affecting AR-V7 positivity on our univariate analysis. PSA (SD) was 529.8 ng/mL (661.5) for Celsee-processed AR-V7-positive samples and 91.7 ng/mL (167.3) for Celsee-processed AR-V7-negative samples (*p* = 0.016).

Among the patients with bone metastasis, the Celsee AR-V7-positive/negative ratio was 3 (7.3%)/27 (65.9%) in patients with an EOD2 classification or lower, and the Celsee AR-V7 positive/negative ratio was 6 (14.6%)/5 (12.2%) in those with EOD3 or higher (*p* = 0.006).

Following our multivariate analysis (Table 7), we found no significant PSA differences between AR-V7-positive and -negative samples (HR = 1.004; 95% CI = 1.000–1.005; *p* = 0.058). The difference in the ratio of patients with EOD2 or lower/patients with EOD3 or higher was confirmed (HR = 0.110; 95% CI= 0.017–0.725; *p* = 0.022).

## 4. Discussion

AR-V7 positivity may vary with the CTC method used to process the samples. False positive results may misguide a patient’s treatment. When interpreting AR-V7 status, the enrichment method used needs to be considered, and AR-V7 should be detected by PCR (reflecting mRNA levels) or by antibody reactions to measure the protein levels.

Benign blood cells have been shown to express AR-V7. To further confirm the expression of ARV7, Marín-Aguilera revealed that CD4 and CD8 T-cells, B lymphocytes, T-natural killer cells, and monocytes isolated from PBMC became positive AR-V7 in PCR detection [16]. Takeuchi et al. performed a semiquantitative PCR from the whole blood of patients with and without prostate cancer (PCa), and concluded that AR-V7 detection in whole blood may not predict the effectiveness of Abi/Enz for patients with CRPC [15]. Todenhöfer evaluated AR-V7 by TaqMan quantitative PCR on whole blood samples collected in PAXgene tubes and found AR-V7 in 25/33 (75.7%) of the control men without PCa and in 21/37 (56.7%) patients with CRPC. Their Ct value threshold between samples from patients without PCa and samples from patients with CRPC revealed the AR-V7-positive patients with CRPCs were resistant to Abi and had poorer prognoses than AR-V7-negative patients [13]. The Ct threshold cannot be used to determine whether the AR-V7 in the samples was derived from CTCs or hematopoietic cells, and the test specificity must be low. Qu extracted mRNA from peripheral blood mononuclear cells from whole blood samples using a Ficoll–Hypaque density separation, and assessed the presence of AR-V7 using a droplet digital PCR to find that more than 95% of 132 patients with CRPC had confirmed AR-V7 expressions [20]. He also divided samples into those with higher and lower AR-V7 transcript numbers and showed that patients with more AR-V7 transcripts had a shorter time to treatment failure in the Enz cohort, but not in the Abi cohort [20]. With regard to overall survival (OS), Qu found no significant differences, but he also reported that the time to treatment failure and OS were significantly shorter in patients with positive PSA and higher AR-V7 expressions than in the others [20]. These three reports suggest that when CTC enrichment is not performed, AR-V7 derived from hematopoietic cells affects the analysis results and makes the AR-V7 interpretation difficult regardless of the PCR detection accuracy.

In the Sciarra review, the mCRPC AR-V7 positive rate was reported at 18.3% (range 17.8–28.8%) [21], and our Celsee AR-V7 positive rate was similar at 22.0%. However, whether the detected AR-V7 is really derived from CTCs cannot be confirmed by the detection rate. As mentioned in the introduction, PCR results may reflect positivity of CTCs and hematopoietic contaminants. In fact, we counted the cell population of our Celsee enrichment with a flow cytometer and found about 50,000 leukocytes contaminating the samples after the back-flushing step. Generally, leukocytes in 4 mL of blood are estimated at 13,200,000–36,000,000; the leukocyte removal rate ranged from 99.6–99.8%, and the resulting CTC concentration was extremely high. Still, that the Celsee enrichment system analyzes cell components other than CTCs is a limitation.

Our AR-V7-positive patients were all resistant to Abi and Enz treatments (the lack of statistical significance is due to the ineffective cases in AR-V7-negative patients). Possible causes for the resistance include lineage plasticity and AR indifference, glucocorticoid receptor activation, AR gains or LBD mutations, alternative AR variants and genomic structural rearrangements, AR enhancer amplification, and the presence of additional oncogenic pathways other than AR-V7 [22]. Also, DTX, CBZ, and Radium223 may be effective and should be considered as alternatives.

The JHU method is known as the AR-V7 PCR detection method after CTC enrichment. CTC enrichment is done with immuno-magnetic beads coated with anti-EpCAM and anti-Her2, and end-product lysates are evaluated on multiplex PCR with AR-V7, AR, PSA, Prostate Specific Membrane Antigen (PSMA), and Epidermal growth factor receptor (EGFR) primers. In the PROPHECY study, Enz and Abi were effective in 6–11% of AR-V7 positive patients [11]. Physically captured leukocytes in the beads may explain these results by reflecting positive-AR-V7 cells other than CTCs.

In the Epic Sciences platform, the effectiveness of Enz and Abi were both 0% in AR-V7-positive patients [11]. This platform covers all nucleated blood cells thawed on slides, and the risk of missing CTCs is extremely low. We wanted to know whether AR-V7 can be accurately evaluated by analyzing protein-level antibody responses. The AR-V7 antibody has been reported to show nonspecific reactions [23], and we confirmed the presence of false positive responses to the anti-AR-V7 antibody (Appendix A
Appendix A). However, in the Epic Sciences platform, in addition to the usual CTC definition (CTC is cytokeratin-positive, CD45-negative, and 4’,6-diamidino-2-phenylindole (DAPI)-positive), captured images were analyzed using an automated algorithm that characterizes each cell using >90 variables. This morphological digital pathology diagnosis may overcome the non-specificity of the AR-V7 antibody reactions.

Another implication of our results is that AR-V7 positive patients have more bone metastases than those with EOD3, and these patients have more CTCs in their blood. A limitation of our system is its inability to assess the CTC counts. Other limitations of our study include the small number of samples studied. We targeted patients undergoing sequential therapy for CRPC and could not assess OS or progression-free survival. However, other reports have shown that AR-V7 negative patients do not usually have AR-V7 re-evaluations, with disease progression after their initial AR-V7 assessment and before the beginning of treatments for CRPC. Nakazawa showed that therapeutic interventions modify the AR-V7 status [24]. Studying how the sequential therapy affects prognosis depending on the AR-V7 status is important, in order to be able to appreciate the true value of AR-V7 assessments.

## 5. Conclusions

AR-V7 false positives can be suppressed with a Celsee CTC enrichment strategy. All AR-V7-positive patients had resistance to Enz and Abi treatments after Celsee enrichment, and treatment with DTX, CBZ, and Radium223 improved their responses. Our results also showed that some AR-V7-negative patients also presented Enz and Abi resistance, and that AR-V7 is predominantly positive in patients with bone metastases greater than EOD3.

## Figures and Tables

**Figure 1 diagnostics-10-00151-f001:**
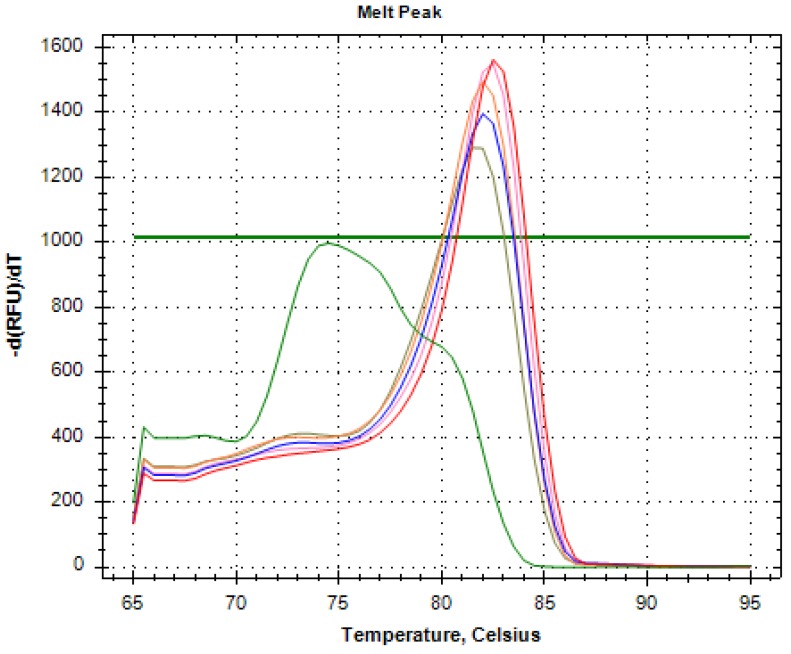
Presence of AR-V7 by PCR in the whole blood of healthy volunteers.

**Figure 2 diagnostics-10-00151-f002:**
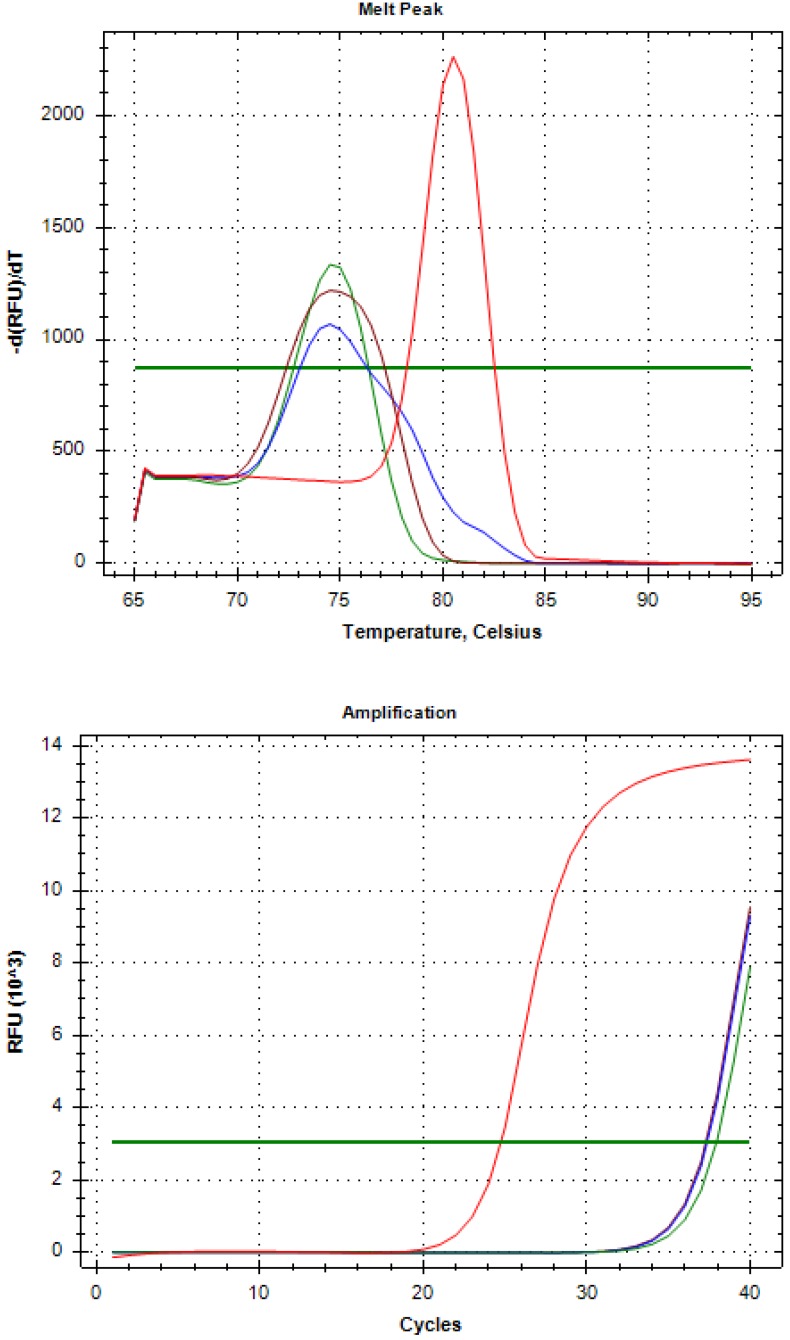
Confirmation of AR-V7 negativity in the whole blood of healthy volunteers after Celsee enrichment.

**Figure 3 diagnostics-10-00151-f003:**
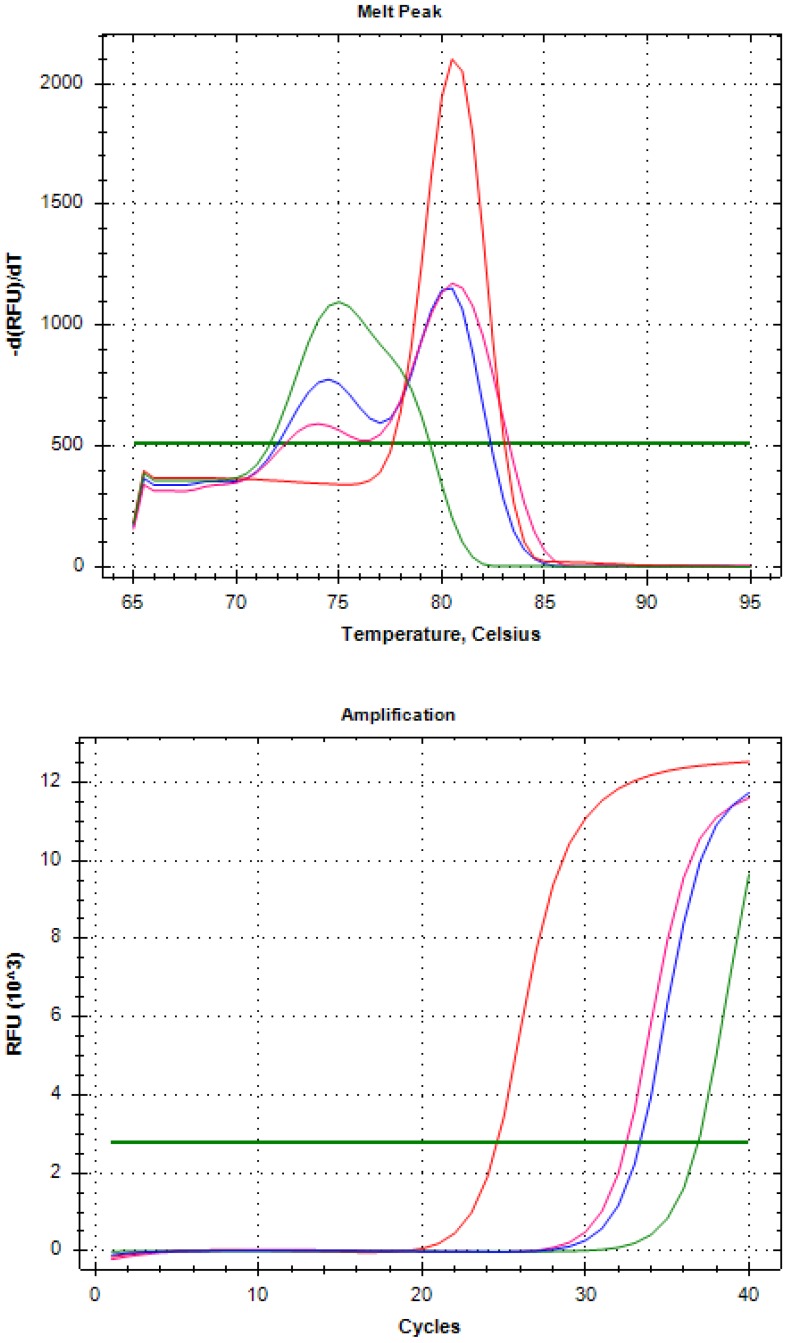
Confirmation of AR-V7 detection on whole blood of healthy volunteers spiked with Vcap (0, 10, 100) after Celsee enrichment.

**Table 1 diagnostics-10-00151-t001:** Baseline characteristics of patients with castration-resistant prostate cancer (CRPC).

Patient Characteristics	*n* (%)	41 (100)
	Mean (standard deviation (SD)) age, years	75.3 (10.3)
	PSA ng/Ml (SD)	187.9 (378.0)
	initial PSA	1437.7 (2898.2)
	Gleason score	*n* (%)
	3 + 3 = 6	3 (7.3)
	3 + 4 = 7	2 (4.9)
	4 + 3 = 7	3 (7.3)
	4 + 4 = 8	9 (21.9)
	4 + 5 = 9	16 (39.0)
	5 + 4 = 9	4 (9.8)
	5 + 5 = 10	4(9.8)
	PS	*n* (%)
	0	8 (19.5)
	1	22 (53.7)
	2	11 (26.8)
	Mean LDH U/L (SD)	260.0 (175.6)
	Mean ALP U/L (SD)	566.3 (771.5)
	Mean Hb g/dl (SD)	11.4 (1.6)
	Mean Alb g/dl (SD)	3.7 (0.6)
	All metastasis	*n* (%)
	None	3 (7.3)
	Yes	38 (92.7)
	Lymph node metastasis	*n* (%)
	None	26 (63.4)
	Yes	15 (36.6)
	Visceral metastasis	*n* (%)
	None	35 (85.4)
	Yes	6 (14.6)
	Bone metastasis	*n* (%)
	0	5 (12.2)
	1 to 5	14 (34.2)
	6 to 19	11 (26.8)
	20 or more	11 (26.8)
	Usage history of Abi and/or Enz	*n* (%)
	None	11 (26.8)
	Yes	30 (73.2)
	Usage history of DTX and/or CBZ	*n* (%)
	None	26 (63.4)
	Yes	15 (36.6)
	Current treatment	*n* (%)
	Abi, Enz	20 (48.8)
	DTX, CBZ	12 (29.3)
	Vintage hormone	8 (19.5)
	Radium223	1 (2.4)
	Effectiveness of Abi or Enz in the past and present	*n* * ^1^ (%)
	Progression disease	21 (70.0)
	Stable or response disease	9 (30.0)
	Effectiveness of current treatment	*n* (%)
	Progression disease	16 (41.0)
	Stable or response disease	23 (59.0)
	Treatment line	*n* (%)
	2	19 (46.4)
	3	16 (39.0)
	4	3 (7.3)
	5	2 (4.9)
	6	1 (2.4)
	Time to CRPC Month (SD)	28.1 (29.7)
	Time from CRPC Month (SD)	27.4 (18.4)
	PAXgene AR-V7	*n* (%)
	Negative	19 (46.3)
	Positive	22 (53.7)
	Celsee AR-V7	*n* (%)
	Negative	32 (78.0)
□	Positive	9 (22.0)

PSA: Prostate-specific antigen, PS: performance status, LDH: lactate dehydrogenase, ALP: alkaline phosphatase, Hb: hemoglobin, Alb: albumin, Abi: abiraterone, Enz: enzalutamide, DTX: docetaxel, CBZ: cabazitaxel, CRPC: castration-resistant prostate cancer. * ^1^: 30 patients (from 41) received Abi or Enz treatments.

**Table 2 diagnostics-10-00151-t002:** Univariate analysis showing factors affecting the effectiveness of the current treatment.

□	□	Stable or Response Disease	Progression Disease	P
Age Years (SD)		73.2 (11.5)	77.9 (7.9)	0.148
Gleason Score, *n* (%)	7 or fewer	6 (14.6)	2 (4.9)	
	8 or more	17 (41.5)	16 (39.0)	0.301
PSA ng/mL (SD)		45.2 (108.9)	370.0 (508.2)	0.016
initial PSA ng/mL (SD)		1351.4 (2814.1)	1548.1 (3080.9)	0.832
LDH U/L (SD)		214.4 (136.4)	318.0 (205.2)	0.074
ALP U/L (SD)		442.3 (594.4)	724.9 (946.5)	0.249
Hb g/dl (SD)		12.1 (1.4)	10.6 (1.5)	0.005
Alb g/dl (SD)		3.8 (0.6)	3.5 (0.6)	0.063
PS, *n* (%)	1 or fewer	19 (46.3)	11 (26.9)	
	2 or more	4 (9.8)	7 (17.0)	0.312
Metastasis, *n* (%)	Yes	22 (53.7)	16 (39.0)	
	None	1 (2.4)	2 (4.9)	0.573
Visceral Metastasis, *n* (%)	Yes	3 (7.3)	3 (7.3)	
	None	20 (48.8)	15 (36.6)	1
Lymph Node Metastasis, *n* (%)	Yes	9 (22.0)	6 (14.6)	
	None	14 (34.1)	12 (29.3)	0.726
Bone Metastasis, *n* (%)	Extent of disease (EOD) 2 or less	18 (43.9)	12 (29.3)	
	EOD3 or more	5 (12.2)	6 (14.6)	0.489
Usage History of Abi and/or Enz, *n* (%)	Yes	16 (38.5)	14 (30.7)	
	None	7 (20.5)	4 (10.3)	0.726
Usage History of DTX and/or CBZ, *n* (%)	Yes	8 (19.5)	7 (17.0)	
	None	15 (36.6)	11 (26.9)	0.786
Current Treatment, *n* (%)	Abi, Enz	10 (24.3)	10 (24.3)	
	DTX, CBZ	8 (19.6)	4 (9.8)	
	Vintage hormone	4 (9.8)	4 (9.8)	
	Radium223	1 (2.4)	0 (0.0)	0.626
Cellsee AR-V7, *n* (%)	Positive	3 (7.3)	6 (14.6)	
	Negative	20 (48.9)	12 (29.2)	0.075
Time to CRPC Month (SD)		36.9 (35.6)	16.7 (13.6)	0.018
Time from CRPC Month (SD)		27.9 (20.2)	26.6 (16.2)	0.819
Treatment Line (SD)	□	2.9 (1.0)	2.7 (0.9)	0.189

PSA: Prostate-specific antigen, PS: performance status, LDH: lactate dehydrogenase, ALP: alkaline phosphatase, Hb: hemoglobin, Alb: albumin, Abi: abiraterone, Enz: enzalutamide, DTX: docetaxel, CBZ: cabazitaxel, CRPC: castration-resistant prostate cancer.

**Table 3 diagnostics-10-00151-t003:** Factors affecting the effectiveness of the current treatment on multivariable analysis.

□	Odd Ratio (95% CI)	***p***
**Time to CRPC Month** **(SD)**	0.958 (0.907–1.012)	0.125
**Hb g/dl** **(SD)**	0.651 (0.374–1.135)	0.13
**PSA ng/Ml** **(** **SD)**	1.004 (0.999–1.009)	0.143

Hb: hemoglobin, CRPC: castration-resistant prostate cancer, PSA: Prostate-specific antigen, CI: confidence interval.

**Table 4 diagnostics-10-00151-t004:** Effectiveness of Enz and Abi in past and present treatments with/without AR-V7.

Celsee AR-V7	Stable or Response Disease	Progression Disease	□
Negative	9	13	
Positive	0	8	0.067

Abi: abiraterone, Enz: enzalutamide.

**Table 5 diagnostics-10-00151-t005:** Effectiveness of DTX, CBZ, and Radium223 with/without AR-V7.

Celsee AR-V7	Stable or Response Disease	Progression Disease	□
Negative	7	1	
Positive	2	3	0.217

DTX: docetaxel, CBZ: cabazitaxel.

**Table 6 diagnostics-10-00151-t006:** Factors affecting AR-V7 positivity on univariable analysis.

□	□	Celsee AR–V7 Positive	Celsee AR-V7 Negative	*p*
**Age (SD)**		74.8 (10.8)	77.1 (8.1)	0.558
**Gleason score, n (%)**	7 or fewer	1 (2.4)	7 (17.1)	
	8 or more	8 (19.5)	25 (61.0)	0.659
**PSA ng/mL (SD)**		529.8 (661.5)	91.7 (167.3	0.041
**initial PSA ng/mL (SD)**		3188.1 (4259.2)	945.5 (2239.9)	0.161
**LDH U/L (SD)**		340.2 (272.5)	237.6 (135.2)	0.302
**ALP U/L (SD)**		1128.4 (1319.5)	408.3 (448.5)	0.143
**Hb g/dl (SD)**		11.0 (1.7)	11.6 (1.6)	0.389
**Alb g/dl (SD)**		3.5 (0.7)	3.7 (0.5)	0.331
**PS, n (%)**	1 or fewer	5 (12.2)	25 (60.9)	
	2 or more	4 (9.8)	7 (17.1)	0.217
**Metastasis, n (%)**	Yes	9 (22.0)	29 (70.7)	
	None	0 (0)	3 (7.3)	1
**Visceral metastasis, n (%)**	Yes	2 (4.9)	4 (9.8)	
	None	7 (17.1)	28 (68.2)	0.597
**Lymph node metastasis, n (%)**	Yes	5 (12.2)	10 (24.4)	
	None	4 (9.8)	22 (53.6)	0.248
**Bone metastasis, n (%)**	Less EOD2	3 (7.3)	27 (65.9)	
	More EOD3	6 (14.6)	5 (12.2)	0.006
**Usage history of Abi and/or Enz, n (%)**	Yes	8 (19.5)	22 (53.7)	
	None	1 (2.4)	10 (24.4)	0.401
**Usage history of DTX and/or CBZ, n (%)**	Yes	5 (12.2)	10 (24.4)	
	None	4 (9.8)	22 (53.6)	0.248
**Current treatment, n (%)**	Abi, enz	2 (4.9)	17 (41.4)	
	DTX, CBZ	4 (9.8)	8 (19.5)	
	Vintage hormone	2 (4.9)	7 (17.1)	
	Radium223	1 (2.4)	0 (0.0)	0.142
**Effectiveness of current treatment, n (%)**	Stable or response disease	3 (7.7)	20 (51.3)	
	Progression disease	6 (15.4)	10 (25.6)	0.075
**Time to CRPC Month (SD)**		11.2 (10.9)	32.5 (31.8)	0.069
**Time from CRPC Month (SD)**		27.7 (13.7)	27.3 (19.7)	0.956
**Treatment Line (SD)**	□	2.8 (1.0)	2.8 (1.0)	0.993

PSA: Prostate-specific antigen, PS: performance status, LDH: lactate dehydrogenase, ALP: alkaline phosphatase, Hb: hemoglobin, Alb: albumin, Abi: abiraterone, Enz: enzalutamide, DTX: docetaxel, CBZ: cabazitaxel, CRPC: castration-resistant prostate cancer.

**Table 7 diagnostics-10-00151-t007:** Factors affecting AR-V7 positivity on multivariable analysis.

□	Odd Ratio (95% CI)	P
PSA	1.003 (1.000–1.005)	0.058
EOD2 or less/EOD3 or more	0.110 (0.017–0.725)	0.022

PSA: Prostate-specific antigen, EOD: Extent of Bone Disease.

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
