# Peer review of "A Novel, Circulating Tumor Cell Enrichment Method Reduces ARv7 False Positivity in Patients with Castration-Resistant Prostate Cancer"

_diagnostics, 2020, doi:10.3390/diagnostics10030151_

Round 1
Reviewer 1 Report
Comments are below:
1. Although expression of AR splice variants in patients with PCa (more frequently in CRPC) have been reported as resistant phenotype induced by AR-directed therapy, changes in the levels of either ARv7 or PSA is no longer considered as prognostic markers. Even, there is no correlation inbetween ARv7 and PSA. These phenotypic changes are currently only being used and monitored to predict therapy selection criteria etc., and therefore, the reviewer concerns about the overall merit of this study unless authors elaborate the rationale.
2. Please explain why ARv7 expression is required for hematopoietic cells. Is it really bad or on the other hand, good in sense to predict therapy responsiveness (e.g. abiraterone acetate or enzalutamide)? Please perform a thorough literature search on AR variant expression in blood cells/ AR regulation of immunity, and elaborate them in the Introduction and Discussion sections.
3. N number is too low to draw conclusion. Please add the limitation of the analysis in the Discussion section.
4. Abbreviation and word redundancy do not seem to be appropriate for the title. You might can rephrase: A novel circulating tumor cell enrichment method reduces ARv7 false positivity in patients with castration-resistant prostate cancer.
Author Response
Answers to the reviewers
Dear reviewer1,
We appreciate all comments of editor and reviewers.
The comments of reviewers have been helpful in allowing us to revise our manuscript.
All the parts we modified this time have been corrected in blue letters.
Answer1:
Certainly, there are reports that reviewers have pointed out, but we do not believe that the possibility that AR-V7 predicts treatment response has been completely ruled out.
We also believe that selecting treatment based on the presence or absence of AR-V7 may improve the prognosis.
Actually in the prospective multicenter validation AR-V7(PRORHECY study), they have prospectively demonstrated that AR-V7 is a strong predictor of clinical outcomes in men with mCRPC treated with abiraterone or enzalutamide. In patients with multiple clinical indicators of poor prognosis, they could identify a significant subset of patients with detectable AR-V7. Knowledge of AR-V7 status may optimize treatment selection beyond clinical measures of prognosis and disease burden.(1)
(1)ArmstrongAJ, HalabiS, LuoJet al.Prospective multicenter validation of androgen receptor splice variant 7 and hormone therapy resistance in high-risk castration-resistant prostate cancer: The PROPHECY study. J Clin Oncol. 2019 May 1;37(13):1120-1129.
Answer2:
AR splicing variants have been identified in healthy human tissues and it has been speculated that the conservation of the AR splicing pattern in different tissues and in evolutionarily distant vertebrate species could indicate the functional importance of these AR forms.(1)
(1)Laurentino S.S., Pinto P.I., Tomas J., Cavaco J.E., Sousa M., Barros A., Power D.M., Canario A.V., Socorro S. Identification of androgen receptor variants in testis from humans and other vertebrates. 2013;45:187–194. doi: 10.1111/j.1439-0272.2012.01333.x.
In a recent report, there have been reports of AR-V7 expression in blood cells. According to them, basal levels of ARV7 in the non-cancer population they collected PBMC from twenty-four non-cancer individuals.ARV7 was detected in 18 of them (75%) .To further confirm the expression of ARV7 in PBMC they studied five subpopulations (CD4 and CD8 T-cells, B lymphocytes, T-natural killer cells [NK], and monocytes) isolated from PBMC of four non-cancer controls.ARV7 mRNA was detected in T-CD4 and B-lymphocytes from two (50% ) controls, in T-CD8 and NK cells from one (25%) individual, and in the monocyte subpopulation from all four (100%) controls.(2)
(2)Androgen Receptor and Its Splicing Variant7 Expression in Peripheral Blood Mononuclear Cells and in Circulating Tumor Cells in Metastatic Castration-Resistant Prostate Cancer.
Marín-Aguilera M, Jiménez N, Reig Ò, Montalbo R, Verma AK, Castellano G, Mengual L, Victoria I, Pereira MV, Milà-Guasch M, García-Recio S, Benítez-Ribas D, Cabezón R, González A, Juan M, Prat A, Mellado B. Cells. 2020 Jan 14;9(1).
We believe that it is very important to remove non-CTC cells in the blood as much as possible and analyze AR-V7 derived from CTC.
Because it suppresses the false positive reaction of AR-V7 from blood cells.
In the Introduction and Discussion, I added a reference and citation for the expression of AR-V7 in normal blood cells.
Answer3:
Following the advice of the viewer, we added a limitation.
Answer4:
Following the reviewer's advice, we changed the title to A novel circulating tumor cell enrichment method reducing ARv7 false positivity in patients with castration-resistant prostate cancer.
Reviewer 2 Report
In this article the authors present a solution to the problem of false positive results when testing for the AR-V7 splice variant in castration-resistant prostate cancer. They processed the test samples using a CTC (circulating tumor cell) enrichment procedure and found that the enrichment reduces the AR-V7 false positive results. The result is interesting and represents an advance in the field. The paper is well-written and I would not ask for any changes. I recommend acceptance as it is.
Author Response
Answers to the reviewer2
Dear reviewer2,
We appreciate all comments of you.
I have revised my article according to the advice of Reviewer 1, and I will attach the revised one.
I am very grateful for your decision. Thank you.
Round 2
Reviewer 1 Report
The authors revised manuscript by addressing the reviewer's concerns and suggestions in a proper manner.